# Comparison of Interview to Questionnaire for Assessment of Eating Disorders after Bariatric Surgery

**DOI:** 10.3390/jcm10061174

**Published:** 2021-03-11

**Authors:** Inbal Globus, Harry R. Kissileff, Jeon D. Hamm, Musya Herzog, James E. Mitchell, Yael Latzer

**Affiliations:** 1School of Public Health, University of Haifa, 199 Aba Khoushy Ave., Mount Carmel, Haifa 3498838, Israel; inbal.balog@gmail.com; 2Maccabi Healthcare Services, Hamered 27, Tel-Aviv 6812509, Israel; 3Mount Sinai Morningside Hospital, 1111 Amsterdam Avenue, New York, NY 10025, USA; Harry.Kissileff2@mountsinai.org (H.R.K.); jeonhamm@gmail.com (J.D.H.); 4Diabetes, Obesity, and Metabolism Institute, Icahn School of Medicine at Mount Sinai, 1 Gustave L. Levy Pl, New York, NY 10029, USA; 5Institute of Human Nutrition, Vagelos College of Physicians & Surgeons, Columbia University, 630 W 168th Street #1512, New York, NY 10032, USA; 6Department of Counseling & Clinical Psychology, Teachers College, Columbia University, 525 W 120th Street, New York, NY 10027, USA; musyaherzog@gmail.com; 7University of North Dakota School of Medicine, 1244 Wildwood Way, Chaska, MN 55318, USA; Mitchelljames7033@gmail.com; 8Faculty of Social Welfare and Health Sciences, University of Haifa, 199 Aba Khoushy Avenue, Mount Carmel, Haifa 3498838, Israel; 9Eating Disorders Institution, Rambam Health Care Campus, HaAliya HaShniya St. 8, Haifa 3109601, Israel

**Keywords:** bariatric surgery, eating disorders, eating pathology, questionnaire, binge eating disorders

## Abstract

The Eating Disorder Examination Interview Bariatric Surgery Version (EDE-BSV) assesses eating pathology after bariatric surgery but requires significant training and time to administer. Consequently, we developed a questionnaire format called the Eating Disorders After Bariatric Surgery Questionnaire (EDABS-Q). This study evaluates the consistency of responsiveness between the two formats. After surgery, 30 patients completed the EDE-BSV and EDABS-Q in a restricted randomized design. Patient reported behavior for each item which was converted to a score following the Eating Disorder Examination-Questionnaire (EDE-Q) scoring scheme. Responses fell into three distributions: (1) dichotomous, (2) ordinal, or (3) unimodal. Distributions of items were not different between the two formats and order did not influence response. Tests of agreement (normal approximation of the binomial test) and association (χ^2^ analyses on binary data and spearman rank order correlations on ordinal items) were performed. Percent concordance was high across items (63–100%). Agreement was significant in 31 of 41 items (Bonferroni-P < 0.001). Association was significant in 10 of 21 in χ^2^–appropriate items (Bonferroni-P < 0.002), and the ordinal items had highly significant correlations between formats (Bonferroni-P < 0.0125). The EDABS-Q is an adequate substitute for the EDE-BSV and may be useful for research and clinical evaluation of eating pathology after bariatric surgery.

## 1. Introduction

### 1.1. Background

Bariatric surgery is effective in limiting the ability of patients to continue a pre-operative pattern of eating large quantities in a short period of time, i.e., to engage in binge eating episodes. However, many patients regain weight post-operation through other manifestations of maladaptive eating, some of which may be disordered [1]. Instruments to assess eating pathology (EP) in this population are limited [2] and fail to discriminate between EP and behaviors to relieve discomfort, such as involuntary vomiting, food intake restriction, increased meal frequency, and excessive chewing before swallowing [3,4,5]. A semi-structured interview to assess post-bariatric EP exists in the modified version of the Eating Disorder Examination (EDE) interview [6], the Eating Disorder Examination-Bariatric Surgery Version (EDE-BSV) [7]. Note: The EDE interview is a psychometrically validated instrument but the EDE-BSV is not. To the best of our knowledge, there is no self-report version of the EDE-BSV interview. An experienced interviewer and a significant amount of time is required to conduct the interview. Consequently, it is desirable to develop a self-report instrument based on this interview that is valid for use in a post-bariatric surgery population.

### 1.2. Objectives

This study describes the development of a self-report questionnaire (from the components of the Eating Disorder Examination-Questionnaire (EDE-Q) [8] and EDE-BSV interview) [7] that assesses eating disorders after bariatric surgery (EDABS-Q) that can potentially be used to assess EP in this population without the need of an interview. The interview was developed by experts in the field of eating disorder and bariatric surgery. They were well familiar with the EDE-Q questionnaire, which is a validated, well-known questionnaire used globally by researchers and clinicians concerned with disordered eating [7]. They published in detail the way they developed the questionnaire and how it was administered in a clinical setting. We used the wording described in their paper, but simply read the same words that were used in the questionnaire we developed from it. We hypothesized that responses from the interview and questionnaire will accord, regardless of the order in which they are administered.

## 2. Materials and Methods

### 2.1. Study Participants

Participant data were collected between August 2018 and June 2019 at several medical centers in Israel. Thirty individuals with a body mass index (BMI) ≥ 35 kg/m^2^, scheduled to undergo one anastomoses gastric bypass surgery (*n* = 8) or sleeve gastrectomy (*n* = 22), were recruited before their scheduled surgery. Patients were wecruited upon arrival to the bariatric surgery committee Screening eligibility criteria were: (1) age: 18–65 years at the start of the study; (2) Jewish individuals from diverse cultural groups, and both sexes; (3) candidates without diagnosis of psychosis taking any antipsychotic medications, or acutely suicidal. We removed candidates with antipsychotic medications as according to the literature usually they do not get an approval for bariatric surgery. 

### 2.2. Description of Interview and Questionnaire

The EDE-BSV consisted of the items that comprise the four symptom categories (or “subscales”) in the EDE-Q [8]: eating concern, weight concern, shape concern, and restraint, however, was adapted in a few ways. Firstly, items that comprised the original restraint category, which is related to pathological behavior, were also extended to a new and revised restraint category, to avoid physical discomfort. Secondly, items pertaining to compensatory behaviors (such as vomiting, spitting and chewing, rumination, and use of laxatives and diuretics), which we have labelled “purging”, were also included as a category. Furthermore, these items were classified into purging to avoid weight gain and purging to avoid physical discomfort. These items and categories were added because the motivation to avoid or mitigate physical discomfort after bariatric surgery could elucidate the endorsement of restriction and compensatory behaviors (Conceição et al., 2013). A self-report version of the EDE-BSV, called the EDABS-Q, was developed from the adapted EDE-BSV. 

The text for all items included in the EDE-BSV and the EDABS-Q (scored and unscored) are in Appendix A (Table A1). Frequencies of behaviors (reported in days, 0–28) were converted into a 7-point ordinal scale according to the following scheme based on EDE-Q [8], which we will refer to as “scaled scores”: 0 days were scored as “0”; 1–5 days were scored as “1”; 6–12 days were scored as “2”; 13–15 days were scored as “3”; 16–22 days were scored as “4”; 23–27 days were scored as “5”; everyday was scored as “6”. 

### 2.3. Study Protocol

The interview and questionnaire were administered in the Hebrew language, at least 1 year after surgery, in a restricted randomized design. Half of the patients received the EDE-BSV interview format first, and the other half, the EDABS-Q questionnaire format first. The EDABS-Q was administered through Qualtrics, and the interview responses were entered into the same Qualtrics program for comparison with the questionnaire. The wording of both the interview and questionnaire was identical. The only difference is that the interviewer read the questions and recorded the answers. The study was carried out in accordance with the latest version of the Declaration of Helsinki. Informed consent was obtained, and the protocol was approved by the Institutional Review Board at Assuta Medical Center, Israel.

### 2.4. Data Analyses

All data analyses were performed in SAS9.4 software (SAS Institute, Inc., Cary, NC, USA). Plots of frequency distributions for responses to each item were examined for uniformity across items. Due to the non-normality of the distribution of item responses, non-parametric analyses were used.

#### 2.4.1. Examination of the Distributions of All Items in the Interview and Questionnaire

Univariate analyses were executed on all items in both formats (EDE-BSV and EDABS-Q). We expected there would be two distinct patterns of response: (1) binary (i.e., only extreme responses, such that predominant response was 0,1 or 5,6) or (2) ordinal, even distribution of responses across the 7-point scale.

In order to determine whether binary or ordinal analysis was appropriate for comparison of formats, we created a plot of the number of participants, out of 30, whose scaled scores (0–6) for each item were in the middle of the range (a score of 2, 3, or 4 out of the 7-point scheme, 0–6). A distribution suitable for ordinal analysis, as opposed to binary, would include roughly the same number of participants across all seven score types (0–6), and because we recruited 30 participants, we considered 4–5 to be the expected count of participants for each of the seven scores  30 participants7 score types=4.5 participants. Consequently, we would expect, assuming some variability, that the mean number of items in the range of 2–4, would be the mathematical product of the number of participants in each score type (i.e., 4.5) ×3 (2, 3, and 4 are three out of seven possible scores). Due to possible variability, the product could range from 9 (3×3) to 15 (5×3). Items with fewer than nine participants in the intermediate range were classified as dichotomous, while items that had nine or more participants in the intermediate range were considered suitable for ordinal analysis.

#### 2.4.2. Analysis Overview 

Two kinds of tests were conducted. “Agreement” (i.e., the proportion of the same responses to an item on the two formats) was assessed by the binomial test. “Association” (i.e., proportion of the same or different response or response pattern in one format compared to the other format) was assessed by means of chi-squared analyses (appropriate for dichotomous or binary data) and Spearman rank-order correlations (appropriate for the ordinal data). Final conclusions from significance tests were determined after the *p*-values were corrected for multiple comparisons by Bonferroni test (uncorrected *p*-value divided by k number of items). Both uncorrected and corrected results are presented. Given the relatively small sample size, conclusions should be considered tentative, although corrected *p*-values are likely to be reproducible. 

#### 2.4.3. Binomial Tests

Significance of agreement of responses on the two formats was measured by means of the normal approximation of binomial test on the probability that response classes (presence or absence of a feature) on each format were identical. If significantly more individuals responded the same on both formats, than expected by chance (for a sample of 30, more than 20, i.e., 67.8%, or 76% with Bonferroni correction), then the two formats agree for that item. The critical value (x) was computed from the sampling distribution of number of agreements on the formats, which is approximately normally distributed (z), by means of the formula $x=z\times \sqrt{Npq}$, where *N* = 30 and *p* and *q* are the probabilities of agreement and non-agreement (Siegel, 1956). Additionally, a normal approximation of the binomial test was conducted on discordant pairs (i.e., positive response on one format but zero on the other) to determine whether the proportion of responses was greater on the questionnaire than interview and conversely.

#### 2.4.4. Chi-Square Analysis

Frequencies of scaled scores (0–6) were grouped into four classes of pairs of scores for the same item in both formats (See Table 1). Pairs were assigned to individuals by the following rules: If scores of an item were zero (i.e., the behavior rated never occurred) for both formats, the individual’s pair was assigned (“0,0”). If scores for both formats were 1 or greater (i.e., positive), the individual’s pair was assigned (“1,1”). If scores for questionnaire were zero but scores for interview were 1 or greater, then the individual’s pair was assigned (“0,1”). If scores for interview were positive, but scores for questionnaire were zero, the individual’s pair was assigned (“1,0”). Concordant pairs are those in which the responses are the same (“0,0” or “1,1”) for the questionnaire and the interview, whereas discordant pairs are those for which the responses are different (“0,1” or “1,0”) between the versions. Thus a 2 × 2 matrix was constructed and chi-squared tests were conducted on the frequencies of these four pairs for each item in order to test whether frequencies of zero and positive responses were the same or different between the two formats. If the instruments are in agreement, the number of concordant pairs should be greater than the number of discordant pairs. The test of these differences is made by χ^2^ on the frequencies of the four possibilities, with the null hypothesis that frequencies in all cells would be the same. The χ^2^ test is the non-parametric equivalent of correlation, and the coefficient of contingency, C (x2N+x2) is analogous to a regression coefficient. A significant *p*-value for chi-square is equivalent to a significant coefficient of contingency and should be interpreted that the two formats agree. The strength of association is given by the coefficient of contingency, which ranges from 0 to 1. The chi-square test is not valid for expected values with frequencies of less than 1 in any of the four cells (Siegel, 1956). During computation of chi-square, the programming was set to report invalid tests, and these are noted in the results (Table 2). The Cohen’s kappa (κ) coefficient was also calculated as another measure of association of responses among all four categories, ranging from completely negatively correlated (κ =− 1.00) through completely random (κ = 0.00) to completely correlated (κ = 1.00). 

#### 2.4.5. Spearman Rank-Order Correlations

For items that met ordinal criteria, Spearman rank order correlations were run on the scaled scores (0–6) from the interview and questionnaire to measure the strength of association between the two formats in a more precise manner than by the Chi-square test.

## 3. Results

### 3.1. Demographics

Participants were mostly female post-bariatric patients (80%). The mean age at the interview was 45.7 y ± 8.5 SD and the mean time post-surgery was 1.8 y ± 0.5 SD. The mean body mass index (BMI) was 40.5 kg/m^2^ ± 4.8 SD before surgery and 28.1 kg/m^2^ ± 5.2 SD at the time of interview.

### 3.2. Distributions and Frequency of Response to Items in the Questionnaire and Interview

Distributions of responses were similar for the two formats, but they did not segregate into two distinct types, and none of them were normal. Figure 1 shows a plot of the sum, or number, of participants that reported in the intermediate range (2, 3, or 4) across all scored items in both formats. As per the ordinal criteria in Section 2.4.1, only two items of all items in the interview (Q81: Dissatisfaction with weight; Q82: Dissatisfaction with shape), and four from the questionnaire (Q79: Shape influenced self-judgement; Q81; Q82; Q83: Uncomfortable seeing body) were classified as ordinal (Figure 1). 

Conversely, almost all responses to purging-related items were zero. In all thirty participants, there were only eight non-zero responses for the fourteen purging-related items (Figure 1). 

Intermediate response frequencies (i.e., responses of 2, 3, or 4), among the remaining dichotomous items, gradually increased across items, with no sharp breaks (Figure 1). Items that comprised the restraint for weight control, restraint to avoid physical discomfort, purging for weight control, and purging to avoid physical discomfort categories were more dichotomous than items that comprise the eating concern, weight concern, and shape concern categories (Figure 1). Furthermore, the items that comprise the eating concern, weight concern, and shape concern categories had higher reported behavior than not, i.e., ten or more patients reported the behavior (Table 1). 

The higher the number of intermediate scores, the more likely that rank order correlations should be performed, which can be determined from both Table 1 (for exact responses to each item) and Figure 1 (for sum of response frequencies in the intermediate range). With the numbers of responses in the intermediate range as a gauge of an ordinal responding pattern (Figure 1), the largest number of ordinal responses was only twelve participants, i.e., less than half the sample, for all items.

### 3.3. Binomial Tests

#### 3.3.1. Tests of Agreement

Before Bonferroni correction (Table 2, binomial on concordant pairs, *p* < 0.05), agreement of responses on the two formats was significant for all items except Q5: Days excluding liked foods to avoid weight gain (*p* > 0.05). After Bonferroni correction (k = 41, α = 0.001), the following ten items were not significant, although their percent concordance was still relatively high (73–76%, Table 2): Q1: Days limiting food to avoid weight gain; Q5, Q6: Days excluding liked foods to avoid physical discomfort; Q8: Follow diet rules to avoid weight gain; Q9: Follow diet rules to avoid physical discomfort; Q76: Proportion of times felt guilty; Q78: weight influenced self-judgement; Q80: Upset if asked to weigh self once/wk for 4 wk, Q82; Q84: Uncomfortable having others see body (*p* > 0.001, Table 2). Given that the confidence interval for the percent concordance is 5.36 ($\sqrt{\left(n=30\right)*\left(p=.50\right)*\left(q=.50\right)}\times 1.96=5.36$), these ten items with slightly lower percent concordances are likely not significantly different. 

#### 3.3.2. Tests on Discordant Pairs (0,1 and 1,0)

After Bonferroni correction (k = 21, α = 0.002), there were no significant differences between the discordant pairs (*p* < 0.002, Table 2). Consequently, there was no significant difference between the number of positive responses on the interview compared to the questionnaire for any item.

### 3.4. Tests of Association

#### 3.4.1. Chi-Squared Tests

Although χ^2^ tests were invalid for twelve items (see Section 2.4.4 for χ^2^ test criteria, Table 2), for the remaining twenty-one items, the χ^2^ was significant for all (C’s: 0.37–0.66, κ’s: 0.39–0.91, *p* > 0.05, Table 2) but one of them (Q5, *p* > 0.05, Table 2). After Bonferroni correction (k = 21, α = 0.002), the non-significant items (*p* > 0.002) from the χ^2^ tests were the same as those in the first binomial tests of agreement (see Section 3.3.1). 

#### 3.4.2. Spearman Rank Order Correlations on Ordinal Items

The four ordinal items (see Section 3.2) were significantly correlated between formats (*p* < 0.05, Figure 2A–H) even with Bonferroni correction (k = 4, α = 0.0125). These items addressed feelings of dissatisfaction with shape and weight, self-judgment, and being uncomfortable with seeing your body. Neither of the slopes were significantly different from 1, nor were the intercepts significantly different from zero.

## 4. Discussion

To the best of our knowledge, this is the first study that aims to assess the consistency in response between the EDE-BSV in interview and questionnaire formats in post-bariatric surgery patients. In addition, an important and novel feature of this comparison that is different from those in the literature [2,8,9,10,11,12,13,14,15,16,17,18] is the careful analysis of every item across the two formats of assessment.

Previous attempts to compare the EDE interview and EDE-Q were conducted among patients with eating disorders [9,10,11,13,15,16,17,18]; in community samples [8,14]; among bariatric candidates and patients [2,12]. The results in these reports did not reveal either the distribution of the responses on individual items, nor the agreement and associations among the items. However, they have combined items in ways that are not fully validated for combinations of non-normally distributed, and particularly, items with a dichotomous distribution. The procedures used in our study relied on distribution-free assumptions. This report, which illustrated consistency of response between the EDE-BSV and EDABS-Q, is the first in a series of planned reports that will further appraise the psychometric quality of the EDABS-Q (reproducibility, construct, criterion, and predictive validity). Thus, further research is needed to complete the findings of the current research.

The current study found that only 2% of all the 30 participants, reported to have any purging-related behavior. These findings are in line with previous studies that found that purging episodes are not a common behavior among this population [2,19]. Most purging-type behaviors after bariatric surgery are related to physical discomfort due to the surgery [20,21]. However, cases of self-induced vomiting related to weight and shape concerns following surgery are rare, and prevalence data have been largely unavailable [7,22]. We might have captured a higher prevalence of purging related to physical discomfort if our sample were closer to the time of the surgery in which this behavior is more likely to occur [20]. On the other hand, perhaps we would have captured a higher prevalence in purging related to weight control (especially those due to “intense exercise”) years after the surgery when weight regain starts and the need to compensate might be higher [19]. 

It should be noted that we also determined specific purging types that are related to post bariatric patients (chewing and spitting and rumination to avoid weight gain or to avoid physical discomfort). Even with a low prevalence of these behaviors, we found a high and significant agreement (93–97%), but because of the low prevalence, these results should be interpreted with caution.

Overall, there was strong agreement between formats across almost every item. Even for the items that did not significantly agree, with highly restrictive criteria, the percentage of responses that agreed on the two formats was close to 75% (50% would be chance). Given the small sample size, and the standard error (=$\sqrt{Npq}$, where N is the number of observations (30) and *p* and q are chance probabilities of same and different responses (=0.5), respectively) of any percentage of ±5.330=17.67%, there is virtually perfect agreement within limits of confidence. 

Previous studies that compared the EDE to the EDE-Q have not reported either the distribution of the responses on individual items, nor the agreement and associations among the items, but rather the relationship between the mean score of each subscales and total scores, which makes it difficult to compare to our results. In addition, the only study that compared the EDE and EDEQ with post-surgery bariatric patients, and not bariatric surgery candidates (before surgery), was de Zwaan et al. (2004). Even though de Zwaan and colleagues did not report associations and agreement between items, they likewise found a high percentage of agreement between subscales (about 90% agreement in eating, weight and shape concern, and 62% in restraint) and found significant correlations in all the subscales. The authors explain the weaker agreement in the restraint subscale by the possibility that subjects might have included the restraint they experienced due to their gastric surgery rather than only the restriction to avoid weight gain. In our study, restraint questions were separated to into two categories (as in the EDE-BSV) in order to clarify it for the subjects. Items of the restraint category highly agreed for most of the items (73–93% and only one item was 63%), and two items that did not reach significance after highly restricted criteria. These items might still need to be modified or rewritten in a clearer way for this population in order to be suitable as a self-report. However, the rest of the items that did not meet criteria for significant agreement or association do not appear to be systematically different in any clinically meaningful way from the other items, and it is most likely that not reaching statistical significance is simply attributable to variability inherent in a small sample.

There are three major strengths: (1) the convenience of a self-report questionnaire for assessment, research and clinical purposes; (2) its significance to a very specific population, i.e., post-bariatric surgical patients; (3) rigorous data analyses appropriate for specific types of questions applicable for this population, which increase confidence that the self-report questionnaire is an adequate substitute of the interview. Consequently, we can also pinpoint which items should be modified for the next iteration of the EDABS-Q. Another strength is that the study took place in Israel, which has a very diverse population, and comprised a variety of socioeconomic classes, cultures, and religions in a high-quality public health system. However, we suggest replication of this study in other populations and countries.

Potential limitations should be considered because of the small sample size. In addition, the time period chosen was between 1 and 2 years following the surgery. This stage after the operation may be too early to observe specific eating pathologies especially among items that address purging to control weight. We have taken the approach of first validating the questionnaire from the interview, because it is more practical to administer the questionnaire to a sufficiently large sample to determine validity of the various items and whether they coalesce into scales and subscales that are similar to the original EDE or different.

## 5. Conclusions

The strong agreement and association between the EDABS-Q and the interview demonstrates that the EDABS-Q is a valid representation of the information obtained in the interview and that it could be used as a substitute for the interview without significant loss of information. There was no evidence from either type of analysis that the interview and questionnaire were systematically different in any way. The EDABS-Q should be a useful alternative for time-consuming clinical interviews for research on post-bariatric surgery patients.

## Figures and Tables

**Figure 1 jcm-10-01174-f001:**
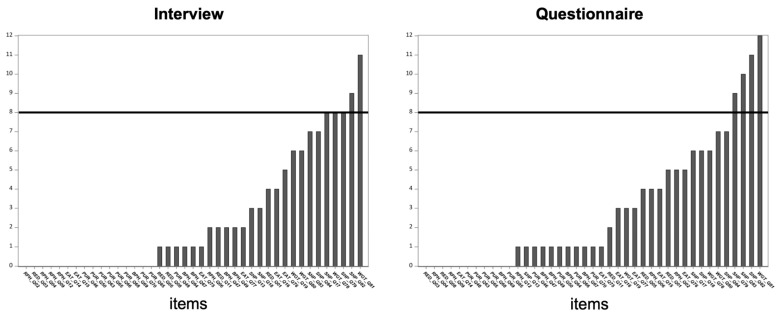
Graphical illustration of intermediate scores across items in the interview and questionnaire. Graph showing the number of participants (n = 30) intermediate scores (between 2 and 4) in the interview and questionnaire. The bold horizontal lines (at the 8 tick on the ordinate axis) are the cut offs between ordinal and not ordinal, with any items with bars breaching this line being considered ordinal. Items on the abscissa are in ascending order, for each format. The first three letters of each item correspond with a behavior/symptom category—RED, restraint for weight control; RPH, restraint to avoid physical discomfort; BPH, purging to avoid physical discomfort; PUR, purging for weight control; EAT, eating concern; WGT, weight concern; SHP, shape concern. See Appendix A for the full text of each item.

**Figure 2 jcm-10-01174-f002:**
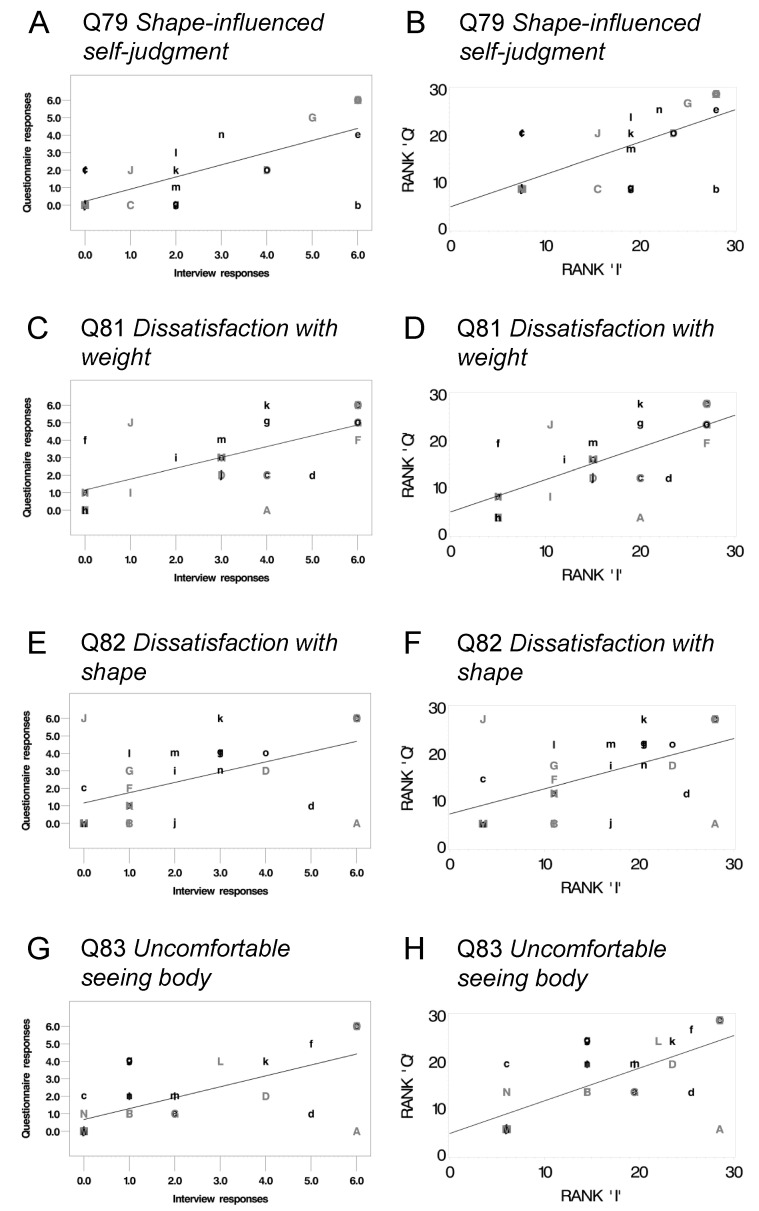
Panel of Pearson linear regression and Spearman rank order correlations plots. (**A**,**C**,**E**,**G**) are Pearson linear regression plots with scaled score response (0–6) from questionnaire (ordinate) regressed from interview (abscissa). Note: Since the distributions of response for these items are not normal, Pearson linear regression is not appropriate, and the presentation of these plots are for visual aid. (**B**) (R^2^ = 0.72, *p* < 0.001), (**D**) (R2 = 0.69, *p* < 0.001), (**F**) (R2 = 0.54, *p* < 0.002), and (**H**) (R2 = 0.70, *p* < 0.001) are Spearman rank order correlation plots with the rank of questionnaire response (ordinate) and the rank of interview response (abscissa). Item number for (**A**–**H**) noted below the title of each plot. All items significant after Bonferroni correction (k = 4, α = 0.0125). See Appendix A for the full text of each item.

**Table 1 jcm-10-01174-t001:** Frequency of ordinal responses to the interview and questionnaire ^a^.

		Interview							Questionnaire						
Category	Item	0	1	2	3	4	5	6	0	1	2	3	4	5	6
*Shape concern*	13. Days desiring flat stomach	13	6	0	2	1	0	8	16	3	0	0	1	1	9
*Shape concern*	15. Difficulty concentrating due to shape or weight concern	24	3	0	0	0	0	3	24	1	2	1	1	0	1
*Shape concern*	17. Fear of weight gain	4	7	4	2	2	0	11	4	5	4	0	2	1	14
*Shape concern*	18. Feeling fat	18	3	0	3	0	0	6	14	3	5	1	0	1	6
*Shape concern*	79. Shape influenced self-judgement	14	2	5	1	2	1	5	16	1	6	1	2	1	3
*Shape concern*	82. Dissatisfaction with shape	6	9	3	4	2	1	5	9	4	2	4	5	0	6
*Shape concern*	83. Uncomfortable seeing body	11	6	4	1	2	2	4	10	6	6	0	4	1	3
*Shape concern*	84. Uncomfortable having others see body	14	4	2	4	1	0	5	13	4	5	0	2	2	4
*Weight concern*	15. Difficulty concentrating due to shape or weight concern	24	3	0	0	0	0	3	24	1	2	1	1	0	1
*Weight concern*	19. Strong weight loss desire	8	4	4	1	1	0	12	6	6	1	2	0	0	15
*Weight concern*	78. Weight influenced self-judgement	12	4	3	2	3	1	5	15	3	5	1	0	2	4
*Weight concern*	80. Upset if asked to weigh self once/wk for 4 wk	9	4	1	4	1	2	9	13	5	3	1	3	1	4
*Weight concern*	81. Dissatisfaction with weight	9	2	1	5	5	1	7	6	3	5	3	4	4	5
*Eating concern*	14. Difficulty concentrating due to food intake	30	0	0	0	0	0	0	30	0	0	0	0	0	0
*Eating concern*	16. Days fearing out of control eating	13	5	3	0	1	1	7	10	5	2	0	1	1	11
*Eating concern*	75. Days eaten in secret	26	3	0	1	0	0	0	25	3	1	0	0	0	1
*Eating concern*	76. Proportion of times felt guilty	14	6	2	3	0	0	5	12	9	2	1	2	2	2
*Eating concern*	77. Concern over other people seeing you eat	27	1	1	1	0	0	0	25	2	1	1	1	0	0
*Restraint for weight control*	1. Days limiting food to avoid weight gain	20	4	2	2	0	1	1	18	4	2	2	1	0	3
*Restraint for weight control*	3. Days w/o eating to avoid weight gain	29	1	0	0	0	0	0	27	3	0	0	0	0	0
*Restraint for weight control*	5. Days excluded liked foods to avoid weight gain	21	3	1	0	0	1	4	18	4	2	2	0	2	2
*Restraint for weight control*	8. Follow diet rules to avoid weight gain	20	2	0	0	1	1	6	13	4	0	0	0	4	9
*Restraint for weight control*	11. Days desiring empty stomach to avoid weight gain	23	3	1	1	0	0	2	21	4	0	1	1	1	2
*Restraint to avoid physical discomfort*	2. Days limiting food to avoid physical discomfort	20	7	0	0	0	0	3	16	7	2	0	3	0	2
*Restraint to avoid physical discomfort*	4. Days w/o eating to avoid physical discomfort	29	1	0	0	0	0	0	28	2	0	0	0	0	0
*Restraint to avoid physical discomfort*	6. Days excluded liked foods to avoid physical discomfort	19	6	1	1	0	0	3	19	6	1	3	0	0	1
*Restraint to avoid physical discomfort*	9. Follow diet rules to avoid physical discomfort	24	0	0	0	0	0	6	18	3	0	0	0	3	6
*Restraint to avoid physical discomfort*	12. Days desiring empty stomach to avoid physical discomfort	23	4	0	0	0	0	3	22	3	1	0	0	1	3
*Purging for weight control*	46. Vomit to lose weight or avoid weight gain	29	1	0	0	0	0	0	29	0	0	1	0	0	0
*Purging for weight control*	48. Chewed food/spit wo swallow to lose weight/avoid weight gain	28	1	0	1	0	0	0	29	1	0	0	0	0	0
*Purging for weight control*	50. Upset after chewing food or spit out w/o swallow	30	0	0	0	0	0	0	29	0	1	0	0	0	0
*Purging for weight control*	63. Days ruminated food to lose weight or avoid gaining weight	30	0	0	0	0	0	0	30	0	0	0	0	0	0
*Purging for weight control*	65. Upset when ruminated food	29	0	0	0	0	0	1	30	0	0	0	0	0	0
*Purging for weight control*	66. Take laxatives to lose weight or avoid gaining weight	30	0	0	0	0	0	0	29	0	1	0	0	0	0
*Purging for weight control*	68. Take diuretics to lose weight or avoid gaining weight	30	0	0	0	0	0	0	30	0	0	0	0	0	0
*Purging for weight control*	70. Driven or compulsive exercise to lose weight	29	1	0	0	0	0	0	26	2	1	0	0	0	1
*Purging for weight control*	85. Vomited in last 4 wks, how upset about it	30	0	0	0	0	0	0	30	0	0	0	0	0	0
*Purging to avoid physical discomfort*	47. Vomit to avoid physical discomfort	25	3	1	0	1	0	0	27	2	0	1	0	0	0
*Purging to avoid physical discomfort*	49. Chewed food/spit w/o swallow to avoid physical discomfort	25	3	0	2	0	0	0	26	3	0	1	0	0	0
*Purging to avoid physical discomfort*	64. Days ruminated food to avoid physical discomfort	27	1	0	0	1	0	1	29	0	0	0	1	0	0
*Purging to avoid physical discomfort*	67. Take laxatives to avoid physical discomfort	27	2	0	0	1	0	0	27	2	1	0	0	0	0
*Purging to avoid physical discomfort*	69. Take diuretics to avoid physical discomfort	30	0	0	0	0	0	0	30	0	0	0	0	0	0

^a^ Table of frequencies for each item, format, and response. N = 30. See Appendix A for full text of questions.

**Table 2 jcm-10-01174-t002:** Comparison of responses to items in the interview and questionnaire.

Category ^d^	Item	0,0	0,1	1,0	1,1	Concordant ^a^ (%)	Binomial ^b^ Con *p*	Binomial ^c^ Dis *p*	χ^2^	χ^2^ *p*	C	κ
*Shape concern*	**13. Days desiring flat stomach**	13 (43%)	3 (10%)	0 (0%)	14 (47%)	90%	<0.0001	0.08	20.07	<0.0001	0.63	0.80
*Shape concern*	**15. Diff conc due to shape or wt concern**	23 (77%)	1 (3%)	1 (3%)	5 (17%)	94%	<0.0001	1.00	18.80	<0.0001	0.62	0.79
*Shape concern*	17. Fear of wt gain	3 (10%)	1 (3%)	1 (3%)	25 (83%)	93%	<0.0001	1.00	15.19	<0.0001	0.58	0.71
*Shape concern*	**18. Feeling fat**	14 (47%)	0 (0%)	4 (13%)	12 (40%)	87%	<0.0001	0.05	17.50	<0.0001	0.61	0.74
*Shape concern*	79. Shape influenced self-judgement	12 (40%)	4 (13%)	2 (7%)	12 (40%)	80%	0.001	0.41	11.06	0.001	0.52	0.60
*Shape concern*	82. Dissat with shape	4 (13%)	5 (17%)	2 (7%)	19 (63%)	76%	0.004	0.26	4.80	0.028	0.37	0.39
*Shape concern*	83. Uncomf seeing body	9 (30%)	1 (3%)	2 (7%)	18 (60%)	90%	<0.0001	0.56	18.37	<0.0001	0.62	0.78
*Shape concern*	**84. Uncomf having others see body**	10 (33%)	3 (10%)	4 (13%)	13 (43%)	76%	0.004	0.71	8.44	0.004	0.47	0.53
*Weight concern*	15. Diff conc due to shape or wt concern	23 (77%)	1 (3%)	1 (3%)	5 (17%)	94%	<0.0001	1.00	18.80	<0.0001	0.62	0.79
*Weight concern*	**19. Strong wt loss desire**	6 (20%)	0 (0%)	2 (7%)	22 (73%)	93%	<0.0001	0.16	20.63	<0.0001	0.64	0.81
*Weight concern*	**78. wt influenced self-judgement**	10 (33%)	5 (17%)	2 (7%)	13 (43%)	76%	0.004	0.26	8.89	0.003	0.48	0.53
*Weight concern*	**80. Upset to weigh self once/**	7 (23%)	6 (20%)	2 (7%)	15 (50%)	73%	0.011	0.16	6.21	0.013	0.41	0.44
*Weight concern*	81. Dissat with wt	5 (17%)	1 (3%)	4 (13%)	20 (67%)	83%	<0.001	0.18	10.16	0.001	0.50	0.56
*Eating concern*	14. Diff conc due to food intake	30 (100%)	0 (0%)	0 (0%)	0 (0%)	100%	<0.0001	NA	NA	NA	NA	NA
*Eating concern*	**16. Days fearing OOC eating**	10 (33%)	0 (0%)	3 (10%)	17 (57%)	90%	<0.0001	0.08	19.62	<0.0001	0.63	0.79
*Eating concern*	75. Days eaten in secret	24 (80%)	1 (3%)	2 (7%)	3 (10%)	90%	<0.0001	0.56	11.31	<0.001	0.52	0.61
*Eating concern*	**76. Proportion of times felt guilty**	9 (30%)	3 (10%)	5 (17%)	13 (43%)	73%	0.011	0.48	6.45	0.011	0.42	0.46
*Eating concern*	77. Concern—other people seeing you e	25 (83%)	0 (0%)	2 (7%)	3 (10%)	93%	<0.0001	0.16	17.00	<0.0001	0.60	0.71
*Restr wt control*	**1. Days limiting food to avoid wt gain**	15 (50%)	3 (10%)	5 (17%)	7 (23%)	73%	0.011	0.48	5.63	0.018	0.40	0.43
*Restr wt control*	3. Days w/o eating to avoid wt gain	27 (90%)	0 (0%)	2 (7%)	1 (3%)	93%	<0.0001	0.16	9.31	0.002	0.49	0.47
*Restr wt control*	**5. Days excl liked foods to avoid gain**	14 (47%)	4 (13%)	7 (23%)	5 (17%)	63%	0.144	0.37	1.30	0.250	0.20	0.20
*Restr wt control*	**8. Follow diet rules to avoid wt gain**	12 (40%)	1 (3%)	8 (27%)	9 (30%)	70%	0.029	0.02	6.79	0.009	0.43	0.43
*Restr wt control*	**11. Days empty stomach to avoid gain**	21 (70%)	0 (0%)	2 (7%)	7 (23%)	93%	<0.0001	0.16	21.30	<0.0001	0.64	0.83
*Restr discomfort*	**2. Days limiting food avoid discomf**	15 (50%)	1 (3%)	5 (17%)	9 (30%)	80%	0.001	0.10	11.32	<0.001	0.52	0.59
*Restr discomfort*	4. Days w/o eating to avoid discomf	27 (90%)	1 (3%)	2 (7%)	0 (0%)	90%	<0.0001	0.56	0.07	0.786	0.05	-0.05
*Restr discomfort*	**6. Days excl lik foods avoid discomf**	15 (50%)	4 (13%)	4 (13%)	7 (23%)	73%	0.011	1.00	5.44	0.020	0.39	0.43
*Restr discomfort*	**9. Follow diet rules to avoid discomf**	17 (57%)	1 (3%)	7 (23%)	5 (17%)	73%	0.011	0.03	5.87	0.015	0.40	0.39
*Restr discomfort*	**12. Days empty stom avoid discomf**	22 (73%)	0 (0%)	1 (3%)	7 (23%)	97%	<0.0001	0.37	25.11	<0.0001	0.66	0.91
*Purge wt control*	46. Vomit to lose wt or avoid wt gain	29 (97%)	0 (0%)	0 (0%)	1 (3%)	100%	<0.0001	NA	30.00	<0.0001	0.71	1.00
*Purge wt control*	48. Chewed food/spit wo swallow	28 (93%)	1 (3%)	0 (0%)	1 (3%)	97%	<0.0001	0.32	14.48	0.0001	0.57	0.65
*Purge wt control*	50. Upset after chewing food or spit out	29 (97%)	0 (0%)	1 (3%)	0 (0%)	97%	<0.0001	NA	NA	NA	NA	NA
*Purge wt control*	63. Days ruminated food to lose wt	30 (100%)	0 (0%)	0 (0%)	0 (0%)	100%	<0.0001	NA	NA	NA	NA	NA
*Purge wt control*	65. Upset when ruminated food	29 (97%)	1 (3%)	0 (0%)	0 (0%)	97%	<0.0001	NA	NA	NA	NA	NA
*Purge wt control*	66. Take laxatives to lose/avoid gaining	29 (97%)	0 (0%)	1 (3%)	0 (0%)	97%	<0.0001	NA	NA	NA	NA	NA
*Purge wt control*	68. Take diuretics to lose/avoid gaining	30 (100%)	0 (0%)	0 (0%)	0 (0%)	100%	<0.0001	NA	NA	NA	NA	NA
*Purge wt control*	70. Driven exercise to lose wt	26 (87%)	0 (0%)	3 (10%)	1(3%)	90%	<0.0001	0.08	6.72	0.010	0.43	0.36
*Purge wt control*	85. Vomited in last 4 wks, upset about	30 (100%)	0 (0%)	0 (0%)	0 (0%)	100%	<0.0001	NA	NA	NA	NA	NA
*Purge avoi discomf*	47. Vomit to avoid phys discomf	25 (83%)	2 (7%)	0 (0%)	3 (10%)	93%	<0.0001	0.16	17.00	<0.0001	0.60	0.71
*Purge avoi discomf*	49. Chewed /spit avoid phys discomf	25 (83%)	1 (3%)	0 (0%)	4(13%)	96%	<0.0001	0.32	23.08	<0.0001	0.66	0.87
*Purge avoi discomf*	64. Days ruminated food	27 (90%)	2 (7%)	0 (0%)	1 (3%)	93%	<0.0001	0.16	9.31	0.002	0.49	0.47
*Purge avoi discomf*	67. Take laxatives avoid phys discomf	27 (90%)	0 (0%)	0 (0%)	3 (10%)	100%	<0.0001	NA	30.00	<0.0001	0.71	1.00
*Purge avoi discomf*	69. Take diuretics avoid phys discomf	30 (100%)	0 (0%)	0 (0%)	0 (0%)	100%	<0.0001	NA	NA	NA	NA	NA

Table of within-subjects comparisons of binary responses between interview and questionnaire formats in bariatric surgery patients (N = 30) for the thirty dichotomous items. Items that are **bolded** are those in which χ^2^ tests are appropriate. χ^2^
*p*-values were held to Bonferonni correction for multiple comparisons (k = 21, α = 0.002). C, coefficient of contingency; κ, Cohen’s kappa statistic. ^a^ Concordant is the sum of percentages of concordant pairs (0,0 and 1,1) from the columns to the left. ^b^ Normal approximation of binomial tests were conducted on frequency of pairs by combining concordant pairs (0,0 and 1,1) to determine if the proportion of observed concordant pairs are due to chance (50%). *p*-values were held to Bonferonni correction for multiple comparisons (k = 41, α = 0.001). Con, concordant pairs. ^c^ Separate normal approximation of binomial tests were conducted on the discordant pairs (0,1, positive on interview and zero on questionnaire vs. 1,0, positive on questionnaire and zero on interview) to determine if participants were more likely to answer higher on the interview or questionnaire. *p*-values were held to Bonferonni correction for multiple comparisons (k = 21, α = 0.002). Dis, discordant pairs. ^d^ Category and item are abbreviated. Full text can be found in Table 1 and Appendix A.

## Data Availability

Data available on request due to restrictions eg privacy or ethical. The data presented in this study are available on request from the corresponding author. The data are not publicly available due to patients privacy.

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
