# Peer review of "Comparison of Interview to Questionnaire for Assessment of Eating Disorders after Bariatric Surgery"

_jcm, 2021, doi:10.3390/jcm10061174_

Round 1
Reviewer 1 Report
This was a well-written manuscript looking at the relationship between the interview and questionnaire version of an assessment of eating disorders after bariatric surgery.
The analysis was presented well, though I think that some of the tables could have been cut, or presented in a more succinct format. The sample size was on the small size and other psychometric methods could have been considered also to compensate for this, such as Rasch analysis. It would be good for the authors to refer to their decision-making in relation to the analytic method.
Reviewer 2 Report
Overall, this paper is a nice contribution to the literature with important research and clinical practice implications. Development and testing of a post-bariatric surgery measure to detect disordered eating pathology is certainly needed. In that positive overall context, I have a few minor comments/suggestions:
1-The authors note that the EDE interview is psychometrically validated but not the EDE-BSV. The way that is presented does beg the question about whether the important next step would have been validating the EDE-BSV before developing a measure based on it to then compared to it.
2-More information is needed about whether the interview and questionnaire were identical. How were interviewers trained to administer the interview - to directly read the questions and offer no additional follow up or explanations strictly? Or as in clinical interviews in practice was there less fidelity to the approach? It may also help to include literature on what would be expected to be different between interview and questionnaire (e.g., less fidelity with interview, higher likelihood of impression bias, etc)
3-Why screen out anyone taking psych meds? It seems that would bias the sample to rule out those who may be prone to eating disorders and other mental health conditions.
4-All racial/ethnic backgrounds? What was the breakdown?
5-Very minor: The EDEQ is never introduced/explained, just the acronym used beginning in the abstract, for unfamiliar readers.
